# Inbreeding in Chinese Fir: Insight into the Rare Self-Fertilizing Event from a Genetic View

**DOI:** 10.3390/genes13112105

**Published:** 2022-11-13

**Authors:** Rong Huang, Weishan Zeng, Houyin Deng, Dehuo Hu, Runhui Wang, Huiquan Zheng

**Affiliations:** 1Guangdong Provincial Key Laboratory of Silviculture, Protection and Utilization, Guangdong Academy of Forestry, Guangzhou 510520, China; 2College of Forestry and Landscape Architecture, South China Agricultural University, Guangzhou 510642, China

**Keywords:** Chinese fir, selfing, genetic diversity, genetic structure

## Abstract

Chinese fir (*Cunninghamia lanceolata* (Lamb.) Hook.) is a fast-growing conifer with great forestation value and prefers outcrossing with high inbreeding depression effect. Previously, we captured a special Chinese fir parent clone named as ‘cx569’ that lacks early inbreeding depression. In view of the fact that very little has been published about the rare self-fertilizing event in Chinese fir from a genetic view, herein, we conduct an SSR-based study on the variation of open- and self-pollinated offspring of this parent to gain a view of the rare self-fertilizing event. The results indicated that genetic diversity of self-pollinated offspring was significantly reduced by half (*H*o: 0.302, vs. 0.595, *p* = 0.001; *H*e: 0.274 vs. 0.512, *p* = 0.002) when compared to an open-pollinated set. Self-pollinated offspring also had significantly positive *F*_IS_ values (*F*_IS_ = 0.057, *p* = 0.034) with a much higher proportion of common allele (20.59% vs. 0), reflecting their heterozygote deficiency. Clustering analysis further indicated a separation of the self- and opened- pollinated groups, implying a natural preference of outcrossing for cx569. However, the cx569 still had 6% acceptance for selfing. When accepted 100% for its own pollen, the cx569 led to a genetically unique selfing group. Additionally, this selfing group seemed to be consistently homozygous at seven particular loci. These findings gave us more genetic clues to gain insight into the rare self-fertilizing event in conifer (Chinese fir).

## 1. Introduction

The reproduction mode of plants determines, to a great extent, their genetic properties and the efficacy of natural selection, and thus potential for adaptive evolution [1,2]. The genetic and ecological consequences of mating systems have long been the subject of considerable concern in evolutionary biology. Selfing, where male and female gametes arise from the same parent, is one of the reproduction modes in seed plants [3,4]. It is considered to have short-term ecological and genetic benefits, including reproductive assurance and inherent transmission advantage [5,6,7]. Nonetheless, selfing inevitably has negative effects, such as pollen discounting and specifically inbreeding depression [8,9]. Thus, plants generate numerous mechanisms (e.g., self-incompatibility, herkogamy dichogamy and dioecy) to prevent selfing and to promote outcrossing, potentially avoiding the negative consequences of selfing [4].

The morphological traits that are expected to influence mating systems are known to be highly dynamic, varying widely among populations or individuals, and have been well documented in some species, such as *Arabidopsis lyrata* [10,11], *Arenaria uniflora* [12], the *Leavenworthia* species [13,14,15] and *Eichhornia Paniculata* [16]. The most frequent evolutionary shift of mating systems is the transition from outcrossing to selfing across the plant kingdom [4,17,18]. From the perspective of genetic consequences, selfing is expected to show reduced effective recombination rates and increased homozygosity, resulting in decreasing genetic diversity relative to outcrossing [19,20]. Given the important effects of mating systems on plant genetic properties, it is of considerable interest to assess the genetic consequences of mating system transition [2,21,22,23]. Up to the current date, the effect of the mating system on genetic properties have focused on angiosperms, however, coniferous trees have received much less attention.

*C. lanceolata* belongs to Cupressaceae, with a common name as Chinese fir, and is a fast-growing coniferous tree with great forestation value [24]. It is naturally widespread from the central to the southeastern zones of China, and has been indicated for cultivation in many countries around the world, such as Vietnam, Brazil, New Zealand, Australia and Canada [25,26,27]. *C*. *lanceolata* is a monoecious species, which produces pollen and ovulate cones in different structures within the same individual (Figure 1). The ovulate cones are distributed in the middle to upper crown, while pollen cones are in the middle to lower crown in a single individual [28,29,30]. Generally, ovulate cones sexually mature earlier than pollen cones within the same plant of *C*. *lanceolata* [31]. Temporal or spatial separation of ovulate and pollen cones are likely to serve as an effective mechanism to encourage outcrossing and prevent selfing in *C. lanceolata* [28,29]. Given this, *C*. *lanceolata* is generally regarded as an outcrossing species. Previous studies indicate that self-fertilization gives rise to significant inbreeding depression in *C*. *lanceolata*, reducing different components of fitness, e.g., seed yield, germination rate and seedling growth [32,33,34]. High inbreeding depression may play a central role to prevent the evolution of self-fertilization in the species [32,33]. Our fieldwork and previous studies [29,31] have shown that the overlapping degree of pollen and ovulate cone maturation vary widely among individuals in *C*. *lanceolata*. Synchronism in pollen and ovulate cone maturation was discovered in a special Chinese fir parent clone, named as ‘cx569’ (Figure 1), providing the opportunity for self-fertilization. More interestingly, the comparation of self- and open-pollinated seedling growth in our previous study showed an absence of early inbreeding depression in cx569 [34]. However, very little has been published about the rare self-fertilizing event in the conifer (Chinese fir) from a genetic view. Simple sequence repeats (SSRs) are extensively and successfully employed in population genetic studies because they are co-dominant, multi-allelic, highly polymorphic and reproducible [35,36,37]. Here, we conduct a comparative study on the genetic properties of self- and open-pollinated offspring of cx569 using 20 SSR markers. The aim of the present study was to reveal the effect of the self-fertilizing event on genetic diversity and structure in Chinese fir.

## 2. Materials and Methods

### 2.1. Plant Material and DNA Extraction

The unpollinated ovulate cones of cx569 were bagged, and pollinated with self-pollen in a 2.5th generation seed orchard of Chinese fir that locates in the Xiaokeng State Forest Farm (Guangdong Province, China; N24°70′, E113°81′, alt. 328–339 m) in February 2018. The mature self-pollinated seeds were then collected in November. Simultaneously, open-pollinated seeds of the cx569 were also collected. All seeds were germinated on filtered water and grown in peat soil mixed with 20% expanded-perlite. Then, 200 seedlings of self- and open-pollination were collected randomly for DNA extraction, respectively.

Total genomic DNA was extracted from the fresh and health leaves with a DNAsecure Plant Kit (TIANGEN, Beijing, China). The DNA purity and concentration were evaluated using 1% agarose gel electrophoresis and a NanoDrop-2000 Spectrophotometer (Wilmington, DE, USA).

### 2.2. SSR Genotyping

The quality of markers and the accuracy of the genotyping data significantly influence the effectiveness and success of SSR. In this study, 20 polymorphic SSR markers, except for SSR6, previously developed were used in this study [38,39]. Attributes of the 20 SSR primer pairs are shown in Appendix A. The polymerase chain reaction (PCR) was conducted in a total volume of 25 µL consisting of 0.5 µL DNA (~50 ng), 0.5 µL forward primer (10 µmol/L), 0.5 µL reverse primer (10 µmol/L), 12.5 µL 2 × Taq Plus PCR MasterMix (TIANGEN, Beijing, China), and 11 µL double distilled water. The PCR program was as follows: initial denaturation at 94 °C for 5 min, followed by 35 cycles at 94 °C for 30 s, 55 °C for 30 s, 72 °C for 30 s and final extension at 72 °C for 10 min. Capillary electrophoresis with fluorescence-labeled SSR marker is a common method that heralded accurate and consistent allele sizing with a high degree of automation and throughput [35,36,37]. Here, the forward primers were labeled with one of the fluorescent dyes ROX, FAM, or HEX at the 5′ end in the polymerase chain reaction assay. The PCR products were then subjected to capillary electrophoresis using an ABI3730xl DNA Analyzer (Applied Biosystems, Carlsbad, CA, USA). Genotypes were determined with Gene-Marker 2.2.0 software (SoftGenetics LLC, State College, PA, USA).

### 2.3. Data Analysis

To evaluate the genetic diversity, the number of alleles (*N*a), the Shannon’s information index (*I*), the observed heterozygosity (*H*o), the expected heterozygosity (*H*e) were analyzed with the GenAlEx 6.5 [40]. Then, an independent *t*-test was used to test whether genetic parameters of self-pollinated offspring were statistically lower than those of open-pollinated group in SPSS Statistics 23 (SPSS Inc., Chicago, IL, USA). Allele frequency was also estimated in GenAlEx 6.5.

Three different approaches were used to assess genetic relationship among offspring. Firstly, a Bayesian model-based cluster analysis was implemented with STRUCTURE v2.3.4 [41] to assign individuals to K genetic clusters. Five independent runs for K values ranging from 1 to 10 were conducted under the admixture model with 100,000 burn-in iterations followed by 1,000,000 iterations for Markov chain Monte Carlo (MCMC). The appropriate K value was determined according to the method of Evanno et al. using Structure Harvester program [42]. The software CLUMPP [43] was used to merge the results of multiple iterations of the corresponding K values. Secondly, Power Marker version 3.25 [44] was used to calculate pairwise Nei’s (1983) DA distance between individuals and generate a genetic distances matrix. The genetic distances matrix was then used as input for clustering analysis by the unweighted pair-group method of averages (UPGMA) and neighbor-joining algorithm (NJ) to generate dendrograms in Power Marker, respectively. MEGA 7.0 [45] was employed to plot and edit the dendrogram. Thirdly, Principal Coordinate Analysis (PCoA) was conducted to visualize the genetic relationship among individuals using GenAlEx 6.5.

## 3. Results and Discussion

### 3.1. Polymorphism Analysis of SSR Markers

The lack of genetic variability and narrow genetic base have significantly harmed the discrimination ability of molecular markers in inbred lines [46]. Thus, the development of specific molecular markers with high-resolution is critical to the accurate and rapid discrimination of inbred lines. In this study, SSR profiles for 20 loci are presented in Appendix A. Among the 20 markers tested, 13 (65%) and 20 (100%) were polymorphic in self- and open-pollinated offspring, respectively (Table 1). A total of 33 and 132 alleles were observed across the 20 markers in self- and open-pollinated offspring, respectively. The number of alleles produced by each primer pair varied from one to two with an average of 1.7 ± 0.5 in self-pollinated offspring, while three to 14 with an average of 6.6 ± 3.3 in open-pollinated offspring. SSR1 and SSR11 had the largest number of alleles (Table 1, Appendix A). The observed heterozygosity (*H*o) ranged among loci from 0 to 0.714 and from 0.065 to 0.930 in self- and open-pollinated offspring, respectively, while the expected heterozygosity (*H*e) ranged from 0 to 0.500 and from 0.064 to 0.798 (Table 1). Among the 20 loci, SSR1 had the highest genetic diversity, while SSR12 harbored the lowest diversity. Most notably, some of the markers including SSR1, SSR2, SSR5, SSR7, SSR11, SSR13, SSR15, SSR17, SSR18, SSR20, SSR21, not only showed high values of genetic parameter in open-pollinated offspring, but also had moderate values in complete selfing offspring (Table 1). Similar to our findings, Duan et al. [47] investigated genetic diversity in 149 Chinese fir using the same 20 SSR markers and found these markers with high *H*o, *H*e and *I* values. Having high genetic parameter values for a marker is a vital indicator that the marker can be successfully used for closely related species/cultivars authentication [48]. Therefore, we propose that these loci can serve as high-resolution makers in estimation of genetic diversity and identification of inbred lines of Chinese fir.

### 3.2. Genetic Diversity of Self- and Open-Pollinated Offspring

Compared with outcrossing, selfing is expected to show reduction of genetic diversity [49,50,51]. In this study, the average values of observed heterozygosity (*H*o: 0.302 ± 0.281 vs. 0.595 ± 0.234, *p* = 0.001), expected heterozygosity (*H*e: 0.274 ± 0.242 vs. 0.512 ± 0.198, *p* = 0.002) and Shannon’s index (*I*: 0.384 ± 0.334 vs. 1.003 ± 0.470, *p* = 0.000) for self-pollinated offspring were significantly lower than those for open-pollinated offspring (Table 1). As theory prediction [19,20], complete selfing results in a reduction of 50% genetic diversity when comparing to open-pollinated offspring in Chinese fir. The reason behind this reduction is that selfing may decrease recombination rates, leading to increased homozygosity and thus reduction of diversity [52,53,54]. Such is the case, which can be draw from the differences in fixation index (*F*_IS_) and allele frequency distribution patterns between the two types of offspring. Positive *F*_IS_ value indicates a deficit of heterozygote, while negative *F*_IS_ value suggests heterozygote excess [55]. Here, the self-pollinated offspring had positive *F*_IS_ values (*F*_IS_ = 0.057) (Table 1), suggesting their heterozygote deficiency. On the contrary, the open-pollinated offspring showed negative *F*_IS_ values (*F*_IS_ = −0.159) (Table 1), indicating they presented an excess of heterozygotes. Moreover, they displayed different distribution patterns of allele frequency as illustrated in Figure 2. For self-pollinated offspring, medium-to-high gene frequency allele (0.5 < gene frequency < 1) accounted for the highest proportion (44.12%), followed by low-to-medium frequency allele (0.5 < gene frequency ≤ 0.5) (29.41%) and common allele (allele frequency = 1) (20.59%). Rare allele (allele frequency ≤ 0.05) accounted for a very low proportion (5.88%). Conversely, rare allele accounted for the highest proportion (53.03%), while common allele (allele frequency = 100%) was absent in open-pollinated offspring. These results suggest that the self-pollinated offspring become more or completely homozygous for most alleles (67.41%) relative to the open-pollinated set, thus resulting in decrease of genetic diversity.

### 3.3. Genetic Structure and Mating System in Clone cx569 of Chinese Fir

Bayesian genetic cluster analysis implemented in STRUCTURE showed that, with K = 2, delta K and log likelihood reached a maximum and minimum value, respectively (Figure 3A,B). All the self-pollinated offspring were assigned to the same genetic cluster, while most open-pollinated offspring were assigned to a second genetic cluster (Figure 3C). The UPGMA dendrogram (Figure 4A) was broadly consistent with the unrooted neighbor-joining (NJ) tree (Figure 4B). The 400 offspring were grouped into two clusters comprising all individuals of self-pollinated offspring and most open-pollinated offspring, respectively (Figure 4). PCoA (Figure 5), an alternative mean of detecting and visualizing the genetic structure, revealed a pattern that was also broadly in line with the partitioning results of the STRUCTURE analysis, UPGMA dendrogram and the NJ tree, that is, self- and open-pollinated offspring were clustered as two genetic groups. Moreover, many alleles that were found in open-pollinated offspring were absent in self-pollinated offspring. For example, only two alleles of SSR1 with size of 336 bp and 350 bp were observed in self-pollinated offspring, while additional twelve alleles were found in open-pollinated offspring (Appendix A). Similarly, SSR11 produced 14 alleles with size ranged from 335 bp to 375 bp in open-pollinated offspring, however, only two of those alleles were detected in self-pollinated offspring (Appendix A). The results indicated that many new alleles were introduced from other parents into offspring of cx569 by inter-individual mating. Overall, these findings suggested a natural preference of outcrossing for cx569.

The mating system of the coniferous trees are characterized as predominantly (~90%) outcrossing [56], especially the species that is temporal asynchrony between pollen release and ovulate cone opening (e.g., *Pinus roxburghii* [57]). On the contrary, synchronization in mature male and female cones lead to selfing in some conifers [58,59]. To our knowledge, there is few literatures about the self-fertilizing event in Chinese fir. In this study, ‘cx569’ shows the overlapping of pollen and ovulate cone maturation, which may lead to self-fertilization. To reveal the self-fertilization event in Chinese fir from a genetic view, genetic relationships between individuals of the self- and open-offspring were analysis. STRUCTURE analysis showed that 13 individuals (1, 6, 9, 10, 13, 16, 18, 19, 24, 28, 29, 38, 193) of open-pollinated offspring were grouped with the self-pollinated ones (Figure 3C). Similarly, 13 open-pollinated offspring (1, 9, 10, 13, 16, 18, 19, 24, 28, 29, 38, 193, 195) were clustered together with self-pollinated offspring in the UPGMA tree (Figure 4A), while an additional six individuals (6, 36, 66, 70, 92, 165) were clustered together with a self-pollinated group in the NJ tree (Figure 4B). The PCoA broadly confirmed the results of the above three analyses, that is, 13 open-pollinated offspring (1, 9, 10, 13, 16, 18, 19, 24, 28, 29, 38, 193, 195) were grouped into a self-pollinated group (Figure 5). These results suggest that the 12 open-pollinated offspring clustered together with self-pollinated offspring in the four clustering results were likely the progenies that were produced by natural self-pollination. The finding suggests that cx569 is predominantly (94%) outcrossing in open-pollinated environment, however, it still had 6% acceptance for selfing.

At the genetic level, a selfing organism is more likely to rapidly create homozygotes from recessive favorable mutations than outcrossers, increasing the overall selection acting on them [19,21,60]. In the complete selfing group, only one genotype was observed at 7 loci, i.e., SSR3, SSR4, SSR8, SSR10, SSR12, SSR14 and SSR19, while three or two genotypes were detected at the remainder of loci (Figure 6). This result indicated that the selfing group seemed to be consistently homozygous at the seven loci. Theoretical studies have predicted that selfing can purge the deleterious recessive mutations and fix recessive favorable mutations, and thus rapidly evolve unless it is counteracted by sufficiently high inbreeding depression [19,21,60]. Considering the lack of early inbreeding depression in cx569, we speculate that these homozygous loci may be the beneficial types that are fixed in the offspring by selfing.

## 4. Conclusions

In the present study, our results revealed that the genetic diversity of self-pollinated offspring was significantly reduced by half relative to the open-pollinated group. Moreover, they also showed significantly positive *F*_IS_ values and a much higher proportion of common allele, suggesting their heterozygote deficiency. The clustering analysis showed that 12 open-pollinated offspring clustered together with self-pollinated offspring were likely the progenies that were produced by self-pollination. The results indicated that cx569 naturally prefer cross-fertilization with an outcrossing rate of 94%, however, it still had 6% acceptance for selfing. When being complete selfing, they were likely to be consistently homozygous at some particular loci (i.e., SSR3, SSR4, SSR8, SSR10, SSR12, SSR14 and SSR19) that is likely the beneficial types. The present study advances our understanding of the rare self-fertilizing event in the conifer (Chinese fir).

## Figures and Tables

**Figure 1 genes-13-02105-f001:**
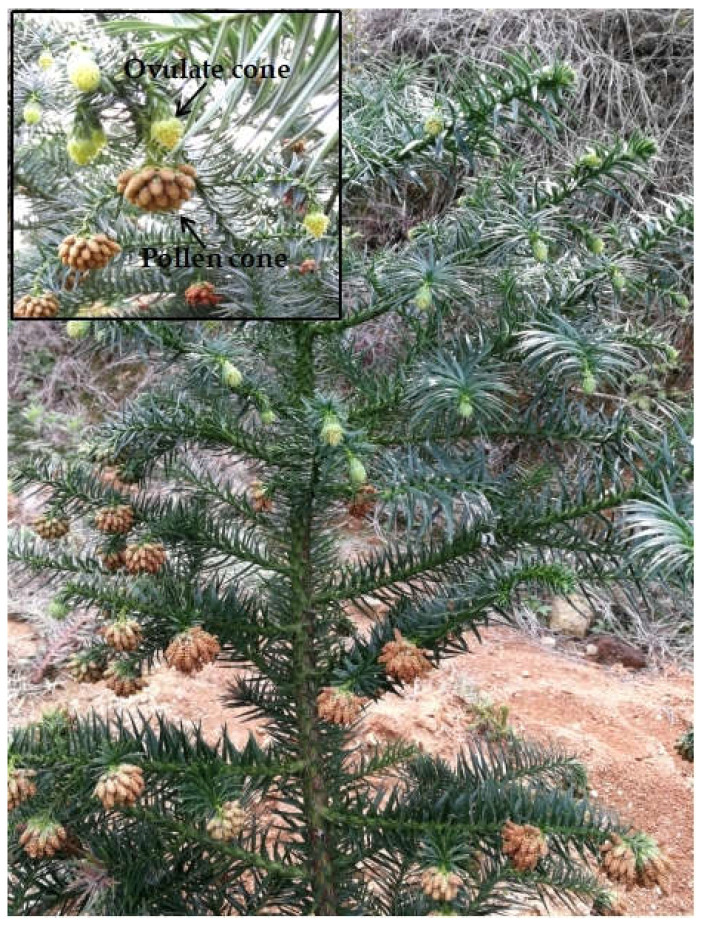
Synchronization in mature of ovulate and pollen cones in clone cx569 of Chinese fir.

**Figure 2 genes-13-02105-f002:**
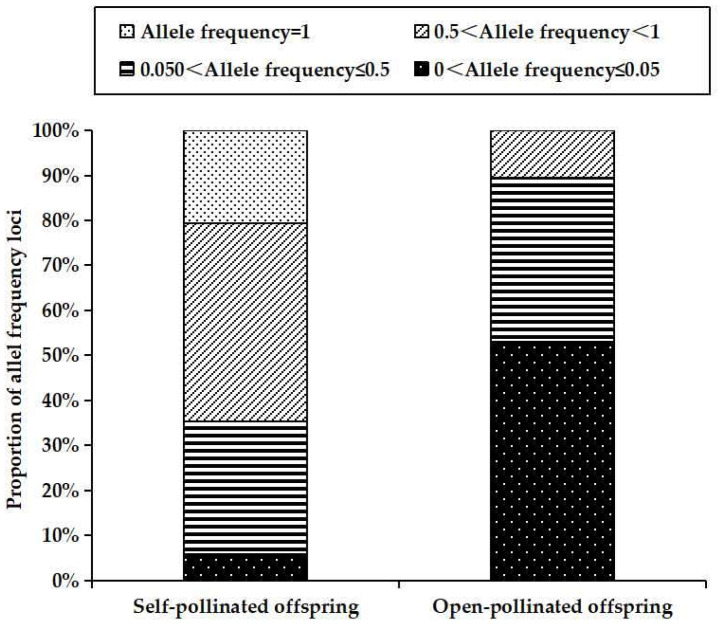
Allele frequency distributions across 20 loci for self- and open-pollinated offspring.

**Figure 3 genes-13-02105-f003:**
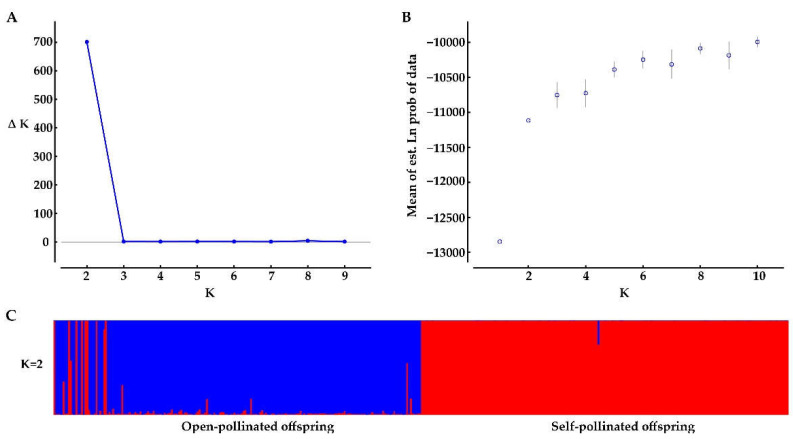
Bayesian clustering results of the STRUCTURE analysis for self- and open-pollinated offspring. (**A**) Estimates of ΔK with respect to K; (**B**) plot of the probability of the data (LnP(D)) values; (**C**) genetic group structure with K = 2, different colors represent different genetic pools.

**Figure 4 genes-13-02105-f004:**
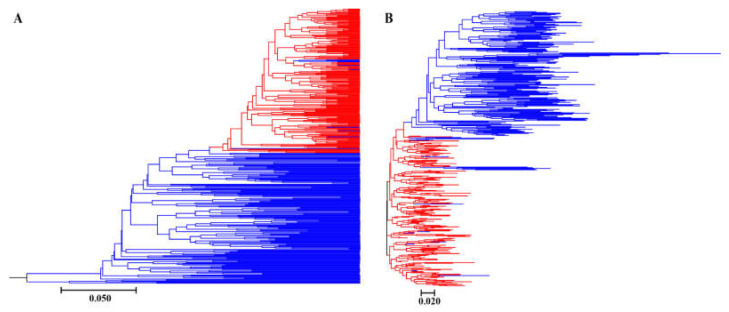
UPGMA dendrogram (**A**) and unrooted neighbor-joining (**B**) trees based on Nei’s (1983) DA distance of self- and open-pollinated offspring. Red lines represent self-pollinated offspring, and blue lines represent open-pollinated offspring.

**Figure 5 genes-13-02105-f005:**
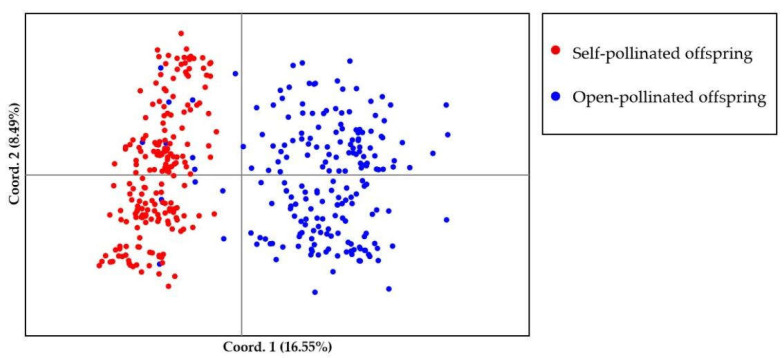
Scatterplot of the principal coordinate analysis (PCoA) for self- and open-pollinated offspring. Different colors represent individuals of self- and open-pollinated offspring.

**Figure 6 genes-13-02105-f006:**
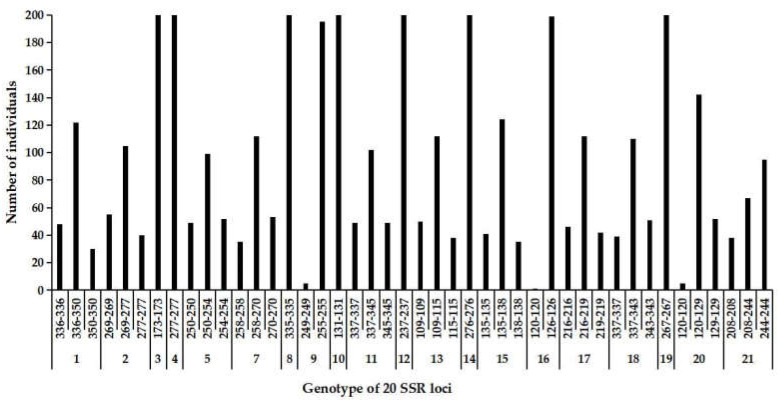
The frequency of genotype across the 20 SSR loci in the self-pollinated offspring. Individual number of a certain genotype was shown in the Y-axis and genotype of SSR loci was show in the X-axis.

**Table 1 genes-13-02105-t001:** Genetic diversity across 20 SSR loci for self- and open-pollinated offspring in clone cx569 of Chinese fir.

Locus	Self-Pollinated Offspring (n = 200)					Open-Pollinated Offspring (n = 200)				
	*N*a	*H*o	*H*e	*I*	*F* _IS_	*N*a	*H*o	*H*e	*I*	*F* _IS_
SSR1	2	0.610	0.496	0.689	−0.230	14	0.865	0.798	1.936	−0.083
SSR2	2	0.525	0.497	0.690	−0.056	12	0.690	0.751	1.709	0.082
SSR3	1	0	0	0	-	5	0.300	0.283	0.588	−0.060
SSR4	1	0	0	0	-	7	0.775	0.567	1.064	−0.366
SSR5	2	0.495	0.500	0.693	0.010	8	0.930	0.737	1.457	−0.263
SSR7	2	0.560	0.496	0.689	−0.129	5	0.570	0.614	1.085	0.072
SSR8	1	0	0	0	-	4	0.195	0.193	0.414	−0.009
SSR9	2	0	0.049	0.117	1.000	6	0.500	0.421	0.844	−0.187
SSR10	1	0	0	0	-	7	0.855	0.568	1.004	−0.505
SSR11	2	0.510	0.500	0.693	−0.020	14	0.885	0.778	1.755	−0.137
SSR12	1	0	0	0	-	5	0.065	0.064	0.181	−0.023
SSR13	2	0.560	0.498	0.691	−0.124	4	0.545	0.518	0.793	−0.053
SSR14	1	0	0	0	-	7	0.565	0.458	0.905	−0.235
SSR15	2	0.620	0.500	0.693	−0.241	5	0.660	0.586	1.056	−0.125
SSR16	2	0	0.010	0.031	1.000	5	0.440	0.375	0.773	−0.174
SSR17	2	0.560	0.500	0.693	−0.120	3	0.435	0.369	0.568	−0.180
SSR18	2	0.550	0.498	0.691	−0.104	5	0.645	0.594	1.016	−0.086
SSR19	1	0	0	0	-	3	0.385	0.329	0.563	−0.169
SSR20	2	0.714	0.472	0.665	−0.511	3	0.770	0.480	0.686	-0.604
SSR21	2	0.335	0.459	0.652	0.271	10	0.815	0.753	1.672	−0.083
Mean	1.7	0.302	0.274	0.384	0.057	6.6	0.595	0.512	1.003	−0.159
SD	0.5	0.281	0.242	0.334	0.435	3.3	0.234	0.198	0.470	0.169

*N*a, the number of alleles; *H*o, the observed heterozygosity; *H*e, the expected heterozygosity; *I*, the Shannon’s information index; *F*_IS_, fixation index; SD, standard error of the mean.

## Data Availability

Not applicable.

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
