# Peer review of "Inbreeding in Chinese Fir: Insight into the Rare Self-Fertilizing Event from a Genetic View"

_genes, 2022, doi:10.3390/genes13112105_

Round 1
Reviewer 1 Report
This study presents an SSR-based study on the variation of open- and self-pollinated offspring of a special Chinese fir parent clone named ‘cx569’ that lacks early inbreeding depression to gain a view of the rare self-fertilizing event.
The results indicated that the genetic diversity of self-pollinated offspring was significantly reduced by half compared to the open-pollinated set.
Abstract:
"Additionally, this selfing group seemed to be consistently homozygous at some particular loci such as SSR3, SSR4, SSR8, SSR10, SSR12, SSR14, and SSR19. "
- I believe that the mention of SSR3, SSR4, etc. in the Abstract is unnecessary, as it is not clear what it is.
In general, the Abstract text is not clear why this research is being done.
96 - Attributes of the 20 SSR primers are shown in Table S1.
- I believe that instead of 20 SSR primers it should be written - 20 SSR primer pairs
It is not clear how the UPGMA dendrogram was prepared in this study
Since the authors showed that open- and self-pollinated offspring differ in allelic variants and their frequencies, with Figures 2 and 3. Therefore, additional Figures 4 and 5 no longer add any new information.
I would propose that the authors provide Polyacrylamide gel electrophoresis (PAGE) figures for some SSR loci that retain polymorphism between open- and self-pollinated offspring.
The authors use the capillary electrophoresis at an ABI3730xl DNA Analyzer, so they can also add data on interesting SSR loci for which a large number of alleles have been identified in the Supplemental form.
In addition, the amount of research in this work is small and consists only of laboratory studies. There is no novelty in the work, and all the results relate only to a special Chinese fir parent clone named 'cx569',
Author Response
Dear reviewer,
We sincerely appreciate your critical reading of our manuscript and helpful comments and suggestions. We have addressed the comments raised by you, and made all changes to the original manuscript using the track-changes feature of Microsoft Word in the revised manuscript. Point by point responses to the comments have been uploaded. We hope the revision alleviates all concerns.
Best regards

Reviewer 2 Report
Dear Authors,
I've received your paper for revision and found it interesting, well written and well structured in the present form.
Kind regards
Author Response
Dear reviewer,
We sincerely appreciate your critical reading of our manuscript and the confirmation of our efforts.
Best regards
Reviewer 3 Report
The paper is focused on the problem of the Chinese fir (Cunninghamia lanceolata (Lamb.) Hook.) which prefers to outcrossing with high inbreeding depression effect . The objective of the study were: SSR-based study on the variation of open- and self-pollinated offspring of this parent to gain a view for the rare self-fertilizing event. The introduction provides a very broad and sufficient background and there were include all relevant references. The research design methods are appropriate to the aim of the study. The methods design was detailed described to be replicated by other researchers. The results are presented clearly, but in my opinion Figure 5 is not necessary, it is reapeated information from figures 3 and 4. The conclusions were derived on the basis of the results. The discussion of the results is very relevant supported by proper literature citations. Nevertheless, in the opinion of the reviewer, the separation of conclusions as a separate more detailed section based on the obtained results would be valuable. This would enable readers to get the conclusions of the research undertaken in the work in a more communicative way.
Author Response

(The authors gave the same response as above.)

Round 2
Reviewer 1 Report
In the period (since 26 September 2022) that the authors have prepared the new version of the paper, I have not found any serious corrections or improvements.
A few lines were added and a few corrections were made.
The authors have prepared Figure S1 (Alleles of SSR1(A) and SSR11 (B) identified by using capillary electrophoresis), during this month, without any description of what it shows.
Authors prefer to leave the work as it is and ignore the reviewer's opinion. I believe that Polyacrylamide gel electrophoresis (PAGE) figures for some SSR loci that retain polymorphism between open- and self-pollinated offspring should be provided. I believe this is a very simple method, and will clearly show what the authors are detecting. Obviously, a full description of the picture should be provided.
There is an overabundance of pictures that do not add information but fill the space of the article. Apparently, it gives more meaning to the results obtained, although the results are only an SSR-PCR analysis.
As I wrote, the work as a whole is a small study, in terms of volume and results. The authors responded to my comments with references to some articles which I will not read. I am expressing my opinion on how I think the work should be.
I think that authors should consider the opinion of reviewers and follow their recommendations.
Author Response
Dear reviewer,
Thank you very much for very helpful and constructive comments. We received the reports from the editor on 24 October, 2022, after the first round of review. We then revised the manuscript according to the comments of reviewers. We are very sorry that the revision did not clarify all concerns. We are now re-submitting our manuscript that has been significantly revised according to these comments and suggestions. Point-by-point response to the comments has been attached as a Word file. Please let us know if additional changes are needed.
Sincerely,
Huiquan Zheng
